# Emergent Zero-Fluoroscopy Mapping and Thoracoscopic Ectomy of Appendage in Pregnant Women with Life-Threatening Atrial Tachycardia: A Case Report and Literature Review

**DOI:** 10.3390/medicina59030528

**Published:** 2023-03-08

**Authors:** Yang Bai, Jie Qiu, Mei Hu, Guangzhi Chen

**Affiliations:** 1Division of Cardiology, Department of Internal Medicine, Tongji Hospital, Tongji Medical College, Huazhong University of Science and Technology, Wuhan 430030, China; 2Health Management Center, Tongji Hospital, Tongji Medical College, Huazhong University of Science and Technology, 1095# Jiefang Ave., Wuhan 430030, China

**Keywords:** pregnancy, zero-fluoroscopy, radiofrequency catheter ablation, atrial tachycardia, multidisciplinary treatment

## Abstract

*Background*: Focal atrial tachycardia (AT) originating from the right atrial appendage (RAA), often persistent and refractory, is clinically rare in pregnant woman, and the therapy is much more challenging. We report that a pregnant woman presented with hypotension due to persistent and refractory atrial tachycardia and was successfully cured by a multidisciplinary treatment (MDT) approach, consisting of a combination of zero-fluoroscopy mapping and thoracoscopic atrial appendectomy. We also carried out a literature review of this topic. *Methods and Results*: A 26-year-old woman in pregnancy at 21 weeks presented with severe palpitation and hypotension due to persistent rapid supraventricular tachycardia (SVT). Since adenosine triphosphate could not terminate the tachycardia, a catheter ablation procedure was planned and finally canceled when the zero-fluoroscopy mapping using Carto 3^TM^ system revealed an atrial tachycardia originating from the RAA. Thoracoscopic RAA ectomy was recommended after multidisciplinary consultation and successfully performed without fluoroscopy. Ensite^TM^ velocity mapping system was used for accurately locating the origin of the arrhythmia during ectomy. The woman finally produced a healthy baby during follow-up. *Conclusions*: Focal AT originating from appendage in pregnant patients can be persistent, refractory, and life-threatening; traditional strategies, such as medicine or catheter ablation, are limited in this situation. MDT measures, using a thoracoscopic ectomy and zero-fluoroscopy three-dimensional electroanatomical mapping technique, is minimally invasive and a promising strategy.

## 1. Introduction

Cardiac arrhythmia in pregnant women is a major therapeutic challenge. Due to a combination of hormonal, hemodynamic, and autonomic changes, an increased incidence of cardiac arrhythmias has been observed in pregnant women [1]. Compared with other arrhythmias, supraventricular tachycardia (SVT) is the most common sustained arrhythmia with a reported incidence of 13 to 24 per 1000 pregnancies [2]. In consideration of the potential risk of radiation exposure to the fetus, catheter ablation is an alternative strategy, whereas antiarrhythmic medicines, including beta-blockers, adenosine triphosphate, and non-dihydropyridine calcium antagonists, used to be the first-choice therapeutic option for pregnant patients with symptomatic SVT [3]. In recent years, however, using real-time intracardiac echocardiographic imaging, three-dimensional (3D) electroanatomical mapping and contact force-sensing catheters, zero-fluoroscopy catheter ablation gradually is preferred in those patients [4].

Focal atrial tachycardias (AT) comprise a distinct subset of SVT and are characterized by centrifugal spread of activation wave fronts from an atrial point source [5]. Using high-density mapping, radiofrequency (RF) catheter ablation can now be effective for curing most focal AT with the three-dimensional mapping technique. However, focal ATs originating from the right atrial appendage (RAA), which constitute an estimated 3.8% of all focal ATs [6,7], may pose big challenges to RF ablation because the atrial appendage (AA) wall is extremely thin and vulnerable. RF ablation here is likely to result in cardiac perforation and a potentially fatal outcome for the mother and fetus. The extremely thin appendage wall can easily become deformed when touched by a catheter, which hinders an accurate location. Additionally, the unusual complex structure can increase the difficulty of catheter ablation.

Thus, it should be a priority to introduce an effective and safe strategy in this situation, especially with continuous and refractory arrhythmia. With limited effect on hemodynamics and tiny wounds, thoracoscopic ectomy of atrial appendage appears to be a prospective method, which only needs to make several apertures at the right third and sixth intercostal spaces for excising the lesion site.

We report a pregnant patient with severe focal atrial tachycardia. The patient successfully underwent thoracoscopic ectomy of the atrial appendage without fluoroscopy using a three-dimensional mapping system.

## 2. Case Presentation

A 26-year-old pregnant woman with rapid and refractory tachycardia, at 21 weeks of gestation, was transferred for urgent management. She had arrhythmia episodes one year before pregnancy, and the arrhythmia became persistent and severe during pregnancy. This patient had a structurally normal heart. The 12-lead electrocardiogram revealed tachycardia of up to 220 beats per minute (Figure 1).

### 2.1. Initial Medication

Adenosine triphosphate administration was given, and then transesophageal overdrive suppression was performed. The tachycardia continued without intermittence and the woman felt dizzy, a little dyspnea, and had mild hypotension.

An immediate and brief multidisciplinary consultation, which comprised electrophysiologists, a cardiac surgeon, an obstetrician, and a medical physicist, agreed that the rapid and persistent tachycardia would be fatal both to the fetus and the mother if it continued. Routinely, the risk of fluoroscopic radiation is often concerned with catheter ablation in pregnancy. It was recommended for the patient since we had carried nearly two thousands cases of zero-fluoroscopy ablation and the success rate of was over 99%.

The patient finally chose to take zero-fluoroscopy catheter ablation using the Carto 3^TM^ system. The study was urgently approved by the Ethics Committee of Tongji Hospital, Tongji Medical College, Huazhong University of Science and Technology, Wuhan, Hubei, China. Written informed consent was acquired.

### 2.2. The First Mapping and Ablation

Antiarrhythmic medications had discontinued for at least five half-lives before the procedure in the primary hospital. Three patches attached to the sensor cables were placed on the back skin of the patient. The location pad was slid under the patient’s table, such that the three back patches were in the center of the location pad. Infiltrated with 1% lidocaine, the femoral vein was punctured. The ablation catheter (NaviStar^®^ 4 mm TC, Biosense Webster Inc., Irvine, CA, USA) was placed via an eight-French sheath; a mapping catheter was placed in the coronary sinus (CS) via a six-French sheath. An electrophysiological study was performed using standard pacing protocols. The entire procedure was performed with zero-fluoroscopy. 

Intracardiac electrogram revealed an SVT at a cycle length of 260 milliseconds. Electrogram from CS electrode showed an activation pattern from proximal to distal. Adenosine triphosphate administration led to an atrium-ventricle conduction ratio of two or three to one. The virtual geometry of the right cardiac chamber was reconstructed by the ablation catheter. Further mapping with the ablation catheter revealed that the tachycardia originated from the apex of right atrial appendage (Figure 2). Ablation with low power, ten to fifteen watts, was applied at the earliest site and had no effect. Higher power and longer ablation strategies were rejected by the expert group due to safety concerns for the pregnant woman and the fetus.

### 2.3. Burst Pacing and Medication

Repetitive burst pacing was applied at three hundred to four hundred beats per minute after the mapping. However, the tachycardia still could not only be terminated but also needed to be converted into atrial fibrillation, a status more applicable to medicine.

Other anti-arrhythmic drugs, including verapamil, propafenone, cedilanid, and esmolol, were administered with the aim of temporally terminating the tachycardia, but they all failed and the tachycardia became more rapid. 

Electric cardioversion was not performed due to the automatic property of the tachycardia.

### 2.4. The Second Mapping and Thoracoscopic Ectomy

The patient was then sent to the Cardiac Care Unit. The tachycardia became more rapid, and her hemodynamics worsened. Hence, another urgent multidisciplinary consultation with the electrophysiologist, cardiac surgeon, obstetrician, and medical physicist was carried out. Plans for additional radiofrequency ablation for the atrial tachycardia were rejected by the expert group. Then, thoracoscopic ectomy of the atrial appendage was recommended and simultaneous three-dimensional mapping was considered accurate for the real-time location.

The patient was sent to the operating room immediately. Three-dimensional mapping guiding by Ensite^TM^ system was performed with zero-fluoroscopy, which also confirmed the same origin. (Figure 3). To avoid surgery, low-power, short-course ablation was performed again, and the results remained ineffective. Attempts to convert atrial fibrillation and terminate tachycardia by high-frequency stimulation also failed. Based on the failure of these attempts, surgical removal of the appendage became inevitable.

Under general anesthesia, after surgical disinfection and surgical drape, with the accessing ports at the right third and sixth intercostal space, the right atrium was on display in front of us (Figure 4). Because of severe volume overload and increased filling pressure, the thin appendage distended and lost contour of its neck, making the right atrium almost like a ball. The surgeon could not stably clamp the appendage through the aperture. Urgent bloodletting, of an estimated 1500 milliliters, had to be performed from the sheath in the femoral vein, which finally helped the surgeon to clamp the shrunken appendage. The blood was infused with an autotransfusion system at last. After excision of RAA by a minimally invasive thoracoscopic approach (Figure 5), her heart rate dropped to 120 beats per minute (bpm) (Figure 6), being confirmed by sinus origin after mapping. The patient received invasive blood pressure monitoring throughout the procedure, which displayed acceptable hemodynamics for the patient and her fetus.

### 2.5. Literature Review

We performed a literature search on PubMed using the following search terms: “pregnancy” AND “atrial tachycardia” AND “catheter ablation”. Studies were reviewed for case reports on catheter ablation being performed in pregnant patients with atrial tachycardia. After screening titles and corresponding content, six published studies met the criteria. 

## 3. Results

### 3.1. Results of Operation

The techniques of catheter introduction, mapping, and ablation were carried out according to established criteria. The patient was able to lie down for the time required to complete the procedure. Catheter mapping was successfully performed on the pregnant woman under the guidance of the Carto 3^TM^ system and Ensite^TM^ system with the zero-fluoroscopy approach. The AT was observed to originate in the apex of the RAA. After a minimally invasive thoracoscopic procedure for RAA excision, the patient restored her sinus rhythm, and the hemodynamics became stable. After surgery, fetal heart monitoring showed that the fetus was in good health. Finally, after several wound dressing changes, the patient’s stitches were successfully removed and she and was discharged.

### 3.2. Follow-Up

This patient delivered a healthy boy by cesarean section. Both mother and child had an uneventful postoperative course. During 6 to 12 months of follow-up, complications related to the ablation and operation were observed neither in the mother nor the child.

### 3.3. Literature Review

A summary of the cases reported previously and our case are shown in Table 1. Seven pregnant patients, undergoing RF catheter ablation for atrial tachycardia, were reported in six published studies [8,9,10,11,12,13]. Focal ATs originating from the AA or other sites were noted in three patients and four patients, respectively. The mean maternal age was 31 years old (range: 20–48), and RF catheter ablation was performed at a mean gestational age of 19.6 weeks (range: 27–30).

All ablations were successfully completed, except one case reported by Mizukami et al. [8]. This was a 26-year-old woman in her first pregnancy presenting with persistent AT, which was also resistant to medication and cardioversion. The first attempt of catheter ablation failed. With the persistent AT, two months after the failed ablation, she developed severe systolic dysfunction and circulatory collapse. The second emergent RF catheter ablation was performed, revealing that the AT originated in the apex of the RAA. Nevertheless, the second attempt of ablation failed again. At last, she also received surgical RAA resection, leading to resolution of the tachycardia and improvement of cardiac function. Pathological examination of the resected RAA revealed a complex AA wall, including diverticulum formation and intramural hematoma, which explained the difficulty of RF catheter ablation. A subsequent pregnancy two years after treatment was uncomplicated.

It is worth noticing that all women and fetuses were in good condition and had an uneventful postoperative course, even for the patient who received surgical RAA resection. However, one patient’s baby, as reported by Yang et al., died during the process and her focal AT also originated from the RAA [11].

## 4. Discussion

The present study reported a pregnant woman presenting with severe, extremely rapid, persistent, and refractory atrial tachycardia originating from RAA. She was suffering from continuous palpitation and hypotension at admission. Initial treatment with adenosine triphosphate failed. Zero-fluoroscopy mapping and ablation was recommended immediately. Three-dimensional mapping revealed an origination of RAA. Then, the planned ablation was canceled for fears of the high risk of cardiac perforation, which might be fatal both to the fetus and woman in this severe situation. Even repetitive burst pacing also could not stop the arrhythmia.

The following measures including repetitive burst pacing and various anti-arrhythmic drugs could not stop the arrhythmia. Electric cardioversion was not performed because of the automatic property of the tachycardia. Thus, thoracoscopic ectomy of the atrial appendage was recommended by an urgent multidisciplinary consultation. Simultaneous three-dimensional mapping was considered for guiding an accurate ectomy.

Of note, the manipulation of the ectomy was quite more difficult in this overweight pregnant woman than in general patients. Furthermore, the thin appendage distended and lost contour of its neck due to severe volume overload and increased filling pressure, making the right atrium almost like a ball; the surgeon could not stably clamp the appendage through the aperture. Urgent bloodletting, of an estimated 1500 milliliters, was performed from the sheath in the femoral vein, which finally helped the surgeon to clamp the shrunken appendage. The blood was infused with an autotransfusion system at last.

### 4.1. Focal AT of Pregnant Women

Pregnant women have a higher risk of arrhythmia than non-pregnant women due to increased blood volume, altered hormone levels, and increased myocardial automaticity [1]. Abnormal automaticity during pregnancy has been found to be the main mechanism for maternal focal AT, and they often manifest around a gestation age of 24–25 weeks. Although recurrent episodes of focal AT may cause tachycardia-induced cardiomyopathy, its prognosis is acceptable with the improvement of left ventricular function and spontaneous resolution of the arrhythmia after delivery in the majority of patients. However, the biological basis for the manifestation of focal AT due to automaticity during pregnancy is still unclear [14].

The underlying reasons may be multi-factorial. First, autonomic adaptations to hemodynamic changes during pregnancy may stimulate some myocardial cells, which have already been vulnerable to arrhythmogenesis. It has been found that these abnormal cells have nodal-like electrophysiological properties with spontaneous automaticity (Phase 4 depolarization) and adenosine sensitivity. Their proarrhythmogenic potential may be unmasked by the increased β-2 adrenergic sensitivity and β adrenergic agonists associated with normal pregnancy. Second, plasma volume markedly increases during normal pregnancy, and the subsequent mechanical stress and stretch of the atrial walls may also facilitate the occurrence of arrhythmias. Third, animal studies found that pregnancy increased the density of the hyperpolarization-activated current (If), which contributed to the normal increase in heart rate in pregnancy but possibly also predisposing for automatic focal AT [14].

### 4.2. Limitation of Conventional Catheter Ablation for Pregnant Women

Despite the safety of medicines and catheter ablation applications, managing pregnant women with SVT has been challenging in clinical practice for a long time [15]. Recently, RF catheter ablation has been recommended as Class IIa for this situation with a level of evidence of C [16]. It is often applied for patients who are suffering from medication failure and experience frequent episodes or present hemodynamic changes. However, conventional RF catheter ablation highly relies on fluoroscopic guidance, which can put both the pregnant women and fetuses at high risk of radiation exposure. Therefore, there is no ideal strategy for curing those arrhythmias in pregnant women [17]. Therefore, it is necessary to develop an effective and safer therapeutic method.

As for conventional catheter ablation, fluoroscopy is necessary for guiding catheter manipulation and visualizing cardiac anatomy, making radiation exposure inevitable. The awareness of radiation doses and risks is essential today in order to apply the risk–benefit assessment and to reinforce the principles of justification and optimization in clinical practice [18]. Radiation exposure related to conventional catheter ablation carries small but non-negligible stochastic and deterministic effects on health, especially for the developing fetus [18]. Andreassi et al. confirmed a definite relationship between long-term exposure in a catheterization laboratory and chromosome breaks [19]. Such radiation exposure is particularly harmful to pregnant women, potentially causing side effects such as fetal death, major organ malformations, intrauterine growth restriction, microcephaly, and cognitive impairment [20]. Previous studies have shown that prenatal radiation at 10 mGy also increases the risk of tumor development in children [21].

### 4.3. Catheter Ablation for Pregnant Women with Zero Fluoroscopy

Advances in electroanatomic mapping (EAM) have made zero-fluoroscopy the goal of many, if not most, catheter ablation procedures. In addition to the reduction in radiation exposure, an EAM system could also offer several benefits. For example, these mapping systems are able to locate the correct position of any electrode at any time and allow an accurate reconstruction of the geometry of both heart chambers and vessels, simplifying navigation and reducing the procedure time [22]. It is also helpful to understand the relationship between bipoles and cardiac anatomy, and between different structures and facilitating complex anatomy cases, continuously visualizing two projections at the same time. EAM systems allow visualization of the catheters from the beginning to the end of the procedure [23].

The advantage of RF catheter ablation utilizing the EAM system has been increasingly acknowledged by healthcare professionals due to the advantage of minimal or even no radiation exposure, making it an ideal choice for pregnant women undergoing catheter ablation [4]. Szumowski et al. reported that nine pregnant women with arrhythmias underwent successful ablation without complications, receiving minimum or no X-ray exposure [4]. Intracardiac echo and electroanatomical mapping systems have also been described to be critical in limiting maternal and fetal radiation exposure [24]. Our center confirmed that the medical staff using the zero-fluoroscopy approach felt less fatigue [25]. We also demonstrated that RF catheter ablation of SVT or PVC/VT in pregnant patients can be safely and effectively performed with a completely zero-fluoroscopy approach [26,27]. Depending on all of these, the European Society for Cardiology recommended that zero-fluoroscopy catheter ablation should be considered in cases of drug-refractory or poorly tolerated SVT in experienced centers [16].

In this case, zero-fluoroscopy mapping and ablation were initially planned since adenosine triphosphate failed to terminate the tachycardia. Carto^®^ system was chosen for the primary mapping because it would be less easy for a cardiac anatomy reconstructed by a magnetic field to become deformed compared with that by an electric field. As a special type of SVT, focal AT is derived from specific sites of the atrium [3,5]. Clinically, the common sites include the crista terminalis, tricuspid annulus, CS ostium, pulmonary veins, mitral annulus, para-Hisian region, and interatrial septum [28,29,30,31,32,33,34,35]. Accurate location of the origin is critical for a successful ablation.

### 4.4. Atrial Tachycardia Originating from Atrial Appendage

Atrial tachycardia originating from AA accounts for 3.8% of the SVT [6,7] and is usually persistent and drug-resistant [36]. It can even cause tachycardia-induced cardiomyopathy [32]. The apex of an atrial appendage is extremely thin and vulnerable, and catheter ablation applied here would be at risk and have higher recurrence despite the use of fluoroscopic guidance; thus, it is often treated by surgical resection [7]. Apart from endocardial access, percutaneous epicardial ablation [37] is an alternative option. However, the reported complications, such as pericardial effusion, phrenic nerve injury, and collateral damage to the coronary artery, make it less favorable [38]. Additionally, further surgery might become more difficult because of the pericarditis when epicardial ablation fails.

Although the atrial appendage possesses endocrine function and can act as a blood buffer to some extent [39], the ectomy appears to have no impact on sinus rhythm, normal cardiac function, and long-term hemodynamic effects, which has been confirmed by previous studies [40]. Ohtsuka et al. proved that removing the atrial appendage was safe [41]. Similarly, left atrial appendage occlusion employed in patients with atrial fibrillation also demonstrated safe outcomes except for low incidences of complications regarding device-related thrombus or catheter manipulation [41,42].

Atrial appendage is relatively complex and comprises thick pectinate muscle bands with intervening thin-walled myocardium, elongated morphology with a multilobulated apex, and a comparatively narrow orifice [7]. Catheter ablation could be far more challenging at the apex of the atrial appendage. About seven years ago, our team evidenced that one woman had cardiac perforation and pericardial perfusion, and the wound had been covered by a thrombus clot due to the catheter ablation, which was performed before the surgical ectomy of appendage. The patient in this study had had severe palpitation, obvious dyspnea, and hemodynamic changes. Prolonged procedure, multiple attempts, and possible recurrence doubtlessly would greatly increase the risk of complications, such as cardiogenic shock or tamponade, which could be fatal both for the mother and the fetus. Therefore, the MDT method, which combined video-assisted thoracoscopic ectomy with three-dimensional and zero-fluoroscopy guidance, a thorough radical measure, was finally taken after consultation. The rapid arrhythmia terminated at once during the clamping and did not occur after the ectomy.

## 5. Conclusions

We presented here a pregnant woman who had severe palpitation, dyspnea, and hypotension due to persistent, refractory, and extremely rapid tachycardia. As adenosine triphosphate could not terminate the arrhythmia, a zero-fluoroscopy catheter ablation was arranged immediately. Three-dimensional mapping with a magnetic field demonstrated an origin from the distal part of RAA. Burst pacing at high rates, ablation with low power, and administration of various drugs, such as verapamil, propafenone, cedilanid, and esmolol, all failed. Considering the risk of ablation using high power and that possible recurrence might be fatal in this situation, video-assisted thoracoscopic ectomy was considered after consultation; three-dimensional electric field mapping was used for accurate location during the ectomy because the X-ray machine in the hybrid operating room was occupied. After 1500 mL of bloodletting, being performed to shrink the bulgy appendage, the surgeon was able to clamp the apex of the appendage. The patient remained stable with autotransfusion and recovered soon after appendage ectomy. She had a healthy baby and had no recurrence during follow-up. MDT measures, using thoracoscopic ectomy and the three-dimensional and zero-fluoroscopy electroanatomical mapping technique, is minimally invasive and a promising strategy.

## Figures and Tables

**Figure 1 medicina-59-00528-f001:**
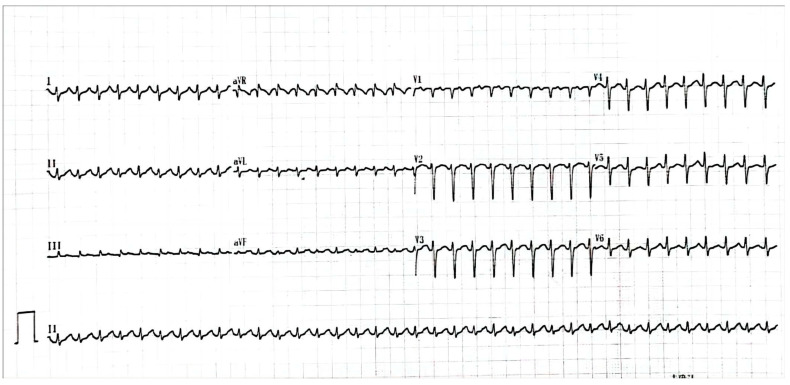
The ECG showed a tachycardia was up to 220 beats per minute.

**Figure 2 medicina-59-00528-f002:**
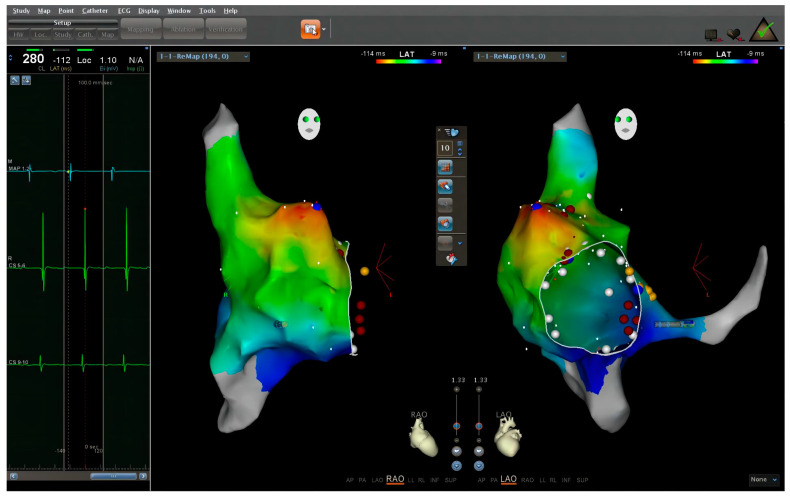
Electrophysiology study was performed with zero-fluoroscopy-approach-guided Carto 3^TM^ system. With the high-density mapping, the originating point was found in the apex of the right atrial appendage.

**Figure 3 medicina-59-00528-f003:**
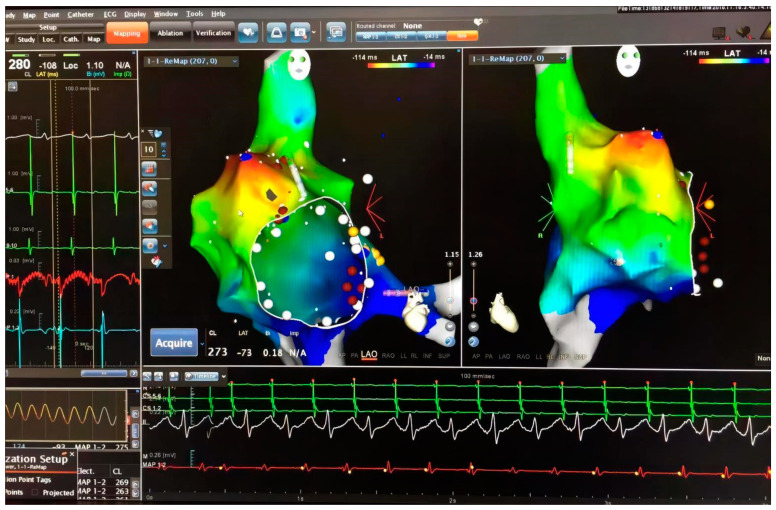
Electrophysiology study was performed again with zero-fluoroscopy approach guided Ensite^TM^ system. With the high-density mapping, the originating point was confirmed in the apex of the right atrial appendage.

**Figure 4 medicina-59-00528-f004:**
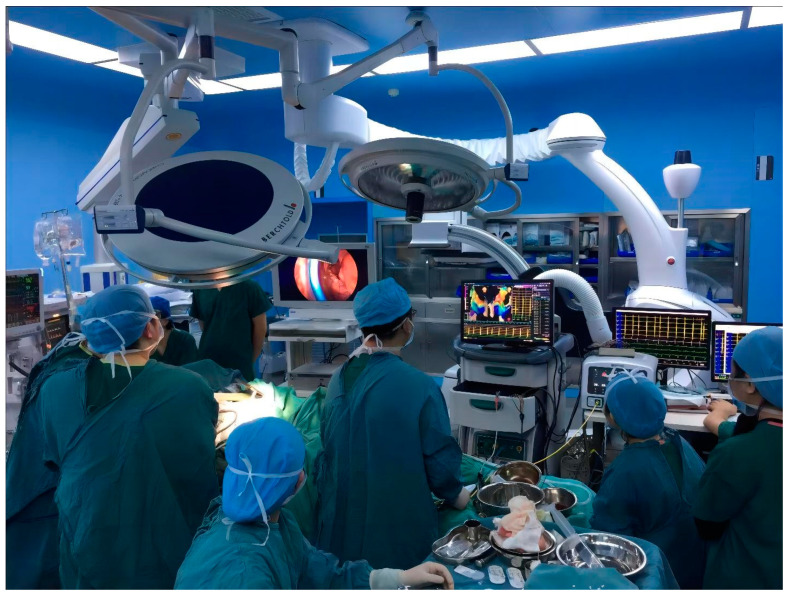
Zero-fluoroscopy mapping and thoracoscopic ectomy of appendage were performed at the same time.

**Figure 5 medicina-59-00528-f005:**
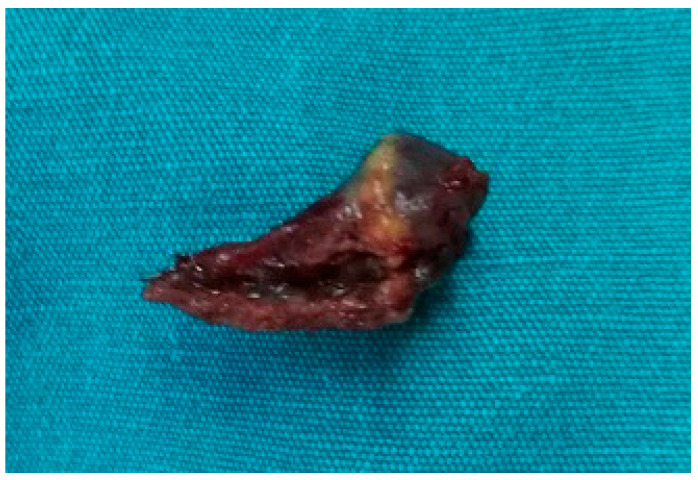
The apex of the right atrial appendage was excised by a thoracoscope operation.

**Figure 6 medicina-59-00528-f006:**
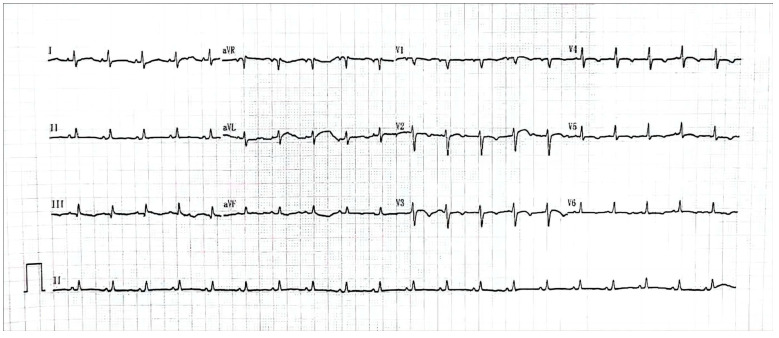
After the operation, this patient’s heart rate returned to 120 bpm.

**Table 1 medicina-59-00528-t001:** Characteristics of previous reported cases and our case.

Author	Origin	Maternal Age (Years Old)	Gestation (Weeks)	Mapping	Results	Refs.
Ferguson (2011)	Not AA	20	27	NavX/ICE	successful	[9]
Zuberi (2014)	Not AA	48	30	NavX	successful	[10]
Yang (2014)	Not AA	30	17	NavX	successful	[11]
	RAA	32	21	NavX	fetal death	[11]
Mizukami (2016)	RAA	26	19	Carto	failed	[8]
Barros (2018)	Not AA	32	12	NavX	successful	[12]
Liu (2019)	LAA	29	11	ICE	successful	[13]
Our case	RAA	26	21	Carto/NavX	successful	

AA = atrial appendage, ICE = intracardiac echocardiography, LAA = left atrial appendage, RAA = right atrial appendage.

## Data Availability

The data is unavailable due to privacy restrictions now.

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
