# Peer review of "Emergent Zero-Fluoroscopy Mapping and Thoracoscopic Ectomy of Appendage in Pregnant Women with Life-Threatening Atrial Tachycardia: A Case Report and Literature Review"

_medicina, 2023, doi:10.3390/medicina59030528_

Round 1

Reviewer 1 Report

Very interesting case report. The authors are to be congratulated for the rare method of treatment.

In my opinion, the article in its current form qualifies for publication.

Author Response

Dear Reviewer: 

Thank you for your comment and recognition of our manuscript entitled “Emergent zero-fluoroscopy mapping and thoracoscopic ectomy of appendage in pregnant women with life-threatening atrial tachycardia: A case report and literature review”. 

Sincerely yours,

   Guangzhi Chen.

Reviewer 2 Report

I would congratulate with authors for the brilliant paper: actually, focal AT originating from appendage in pregnant patients can be persistent and refractory. Conclusions show that thoracoscopic ectomy and zero-fluoroscopy threedimensional electroanatomical mapping technique may be a minimally invasive and a promising strategy. The paper is well written. Here you find only minor comments in order to improve the manuscript in Discussion Section:

Authors should more discuss about mechanisms in maternal focal atrial tachycardia during pregnancy and cite the paper from Wang et (al DOI: 10.1111/jce.14738) concluding that automaticity is the dominant mechanism for patients with maternal focal AT during pregnancy. This aspect and mechanism’s knowledge may impact on the arrhythmia approach. Please clarify this important point and cite suggested reference

We agree that traditional strategy, such as catheter ablation, is potentially limited in this situation. However authors in discussion section should more discuss on every aspect of the new era in Zero X-ray ablation, since the modern technique determines not only potential clinical benefits in terms of reduction of ionising radiation exposure but also  safe technical advantages. In particular contact force catheter, that may be not only a therapeutic approach to arrhythmias, but also a tool for achieving accurate characterization of the arrhythmic substrate in a focal AT in pregnant patient: effective and stable contact between the catheter tip and the tissue is crucial for both mapping and lesion formation during cardiac ablation procedures. (please cite DOI: 10.15420/aer.2020.02).

Another fundamental issue that authors should more explain is that today, the non-fluoroscopic approach is considered a milestone for cancer prevention in ablation procedures, specially in pregnant patients (please cite DOI: 10.1080/00015385.2020.1733303). The awareness of radiation doses and risks, also during interventional cardiology procedures, is essential today in order to apply the risk-benefit assessment and to reinforce the principles of justification and optimisation in clinical practice. (DOI: 10.1007/s10554-011-9937-8) Author should more discuss this important point as well as cite 2 suggested references

Reference list: please update all 4 suggested references

Author Response

Dear Reviewer: 

Thank you for your comments concerning our manuscript entitled “Emergent zero-fluoroscopy mapping and thoracoscopic ectomy of appendage in pregnant women with life-threatening atrial tachycardia: A case report and literature review” . Those comments are valuable and very helpful for revising and improving our paper. We have studied comments carefully and have made correction which we hope meet with approval. The main corrections in the paper and the responds to the reviewer’s comments are as following: 

Responds to the reviewer’s comments: 

1.Response to comment: “Authors should more discuss about mechanisms in maternal focal atrial tachycardia during pregnancy and ...... Please clarify this important point and cite suggested reference”

Response: We have added related content about mechanisms in maternal focal atrial tachycardia during pregnancy, and cited the required references (mark in red).

2.Response to comment:We agree that traditional strategy, ...... and lesion formation during cardiac ablation procedures. (please cite DOI: 10.15420/aer.2020.02).

Response: We have added related content of the required references (mark in red).

3.Response to comment:Another fundamental issue that authors should more explain is that today, ...... Author should more discuss this important point as well as cite 2 suggested references

Response: We have added related content of the required references (mark in red).

 We appreciate for the Reviewer’s warm work, and hope that the correction will meet with approval. Once again, thank you very much for you comments and suggestions.

   Sincerely yours,

Guangzhi Chen.

Reviewer 3 Report

This is very interesting case study showing effective electrophysiology procedure in pregnant women.

Authors presented all details regading clinical status, indications for EP (recurrent, rapid SVT),  technical aspects of the procedure, well-documented EP mapps / Ensite study. Discussion is well prepared with citations of other authors experience (see table). They added data on the long term observation of their patients (healthy born baby).

I have no concerns regarding the manuscript. I recommend the manusript for publication.

Author Response

(The authors gave the same response as above.)

Round 2

Reviewer 2 Report

Manuscript definitely improved respecting all reviewer's suggestions

Congratulations to the authors